# Two-Stage Hybrid Supervision Framework for Fast, Low-resource, and Accurate Organ and Pan-cancer Segmentation in Abdomen CT

Wentao Liu[1][0000−0003−0837−5555], Tong Tian[2][0009−0009−0039−244X], Weijin Xu[1][0000−0001−8371−8330], Lemeng Wang[1][0009−0003−7043−4023], Haoyuan Li[1][0009−0002−1176−3583], and Huihua Yang[1,3][0000−0001−6334−4044]

[1] School of Artificial Intelligence, Beijing University of Posts and Telecommunications, Beijing 100876, China
[2] State Key Laboratory of Structural Analysis, Optimization and CAE Software for Industrial Equipment, School of Aeronautics and Astronautics, Dalian University of Technology, Dalian 116024, China
[3] School of Computer Science and Information Security, Guilin University of Electronic Technology, Guilin, China
liuwentao@bupt.edu.cn

**Abstract.** Abdominal organ and tumour segmentation has many important clinical applications, such as organ quantification, surgical planning, and disease diagnosis. However, manual assessment is inherently subjective with considerable inter- and intra-expert variability. In the paper, we propose a hybrid supervised framework, StMt, that integrates self-training and mean teacher for the segmentation of abdominal organs and tumors using partially labeled and unlabeled data. We introduce a two-stage segmentation pipeline and whole-volume-based input strategy to maximize segmentation accuracy while meeting the requirements of inference time and GPU memory usage. Experiments on the testing set of FLARE2023 demonstrate that our method achieves excellent segmentation performance as well as fast and low-resource model inference. Our method achieved an average DSC score of 90.48% and 50.00% for the organs and lesions on the testing set and the average running time and area under GPU memory-time cure are 11.1 seconds and 8979 MB, respectively.

**Keywords:** Abdominal organ segmentation · Pan-cancer segmentation · Self-training · Mean teacher.

## 1 Introduction

Abdomen organs are quite common cancer sites, such as colorectal cancer and pancreatic cancer, which are the 2nd and 3rd most common cause of cancer death [22]. Computed Tomography (CT) scanning provides important prognostic information for cancer patients and is a widely used technology for treatment monitoring. In both clinical trials and daily clinical practice, radiologists and

clinicians measure the tumor and organ on CT scans based on manual two-dimensional measurements (e.g., Response Evaluation Criteria In Solid Tumors (RECIST) criteria) [5]. However, this manual assessment is inherently subjective with considerable inter- and intra-expert variability. Moreover, existing challenges mainly focus on one type of tumor (e.g., liver cancer, kidney cancer). There are still no general and publicly available models for universal abdominal organ and cancer segmentation at present.

The organizer of FLARE2022 curated a large-scale and diverse abdomen CT dataset, including 4000+ 3D CT scans from 30+ medical centers where 2200 cases have partial labels and 1800 cases are unlabeled. The challenge task is to segment 13 organs (liver, spleen, pancreas, right kidney, left kidney, stomach, gallbladder, esophagus, aorta, inferior vena cava, right adrenal gland, left adrenal gland, and duodenum) and one tumor class with all kinds of cancer types (such as liver cancer, kidney cancer, stomach cancer, pancreas cancer, colon cancer) in abdominal CT scans. Typically, semi-supervised segmentation (SSS) can be employed to resolve this issue. SSS aims to explore tremendous unlabeled data with supervision from limited labeled data. Recently, self-training methods [26,17] have dominated this field. Furthermore, methods employing consistency regularization strategies [26,3,19] improve the generalization ability by encouraging high similarity in predictions from two perturbed networks for the same input image.

In this challenge, due to the fact that the annotation data only includes annotations for partial organs or tumors, traditional SSS methods struggle to achieve excellent segmentation results. The key to developing segmentation algorithms lies in fully leveraging the semantic representation in partially labeled data and extending it to unlabeled cases to enhance the algorithm's generalization. Segmentation of multiple organs and tumors is a generally recognized difficulty in medical image analysis [28], particularly when there is no large-scale fully labeled datasets. To address this issue, [6,21] formulate the partially labeled issue as a multi-class segmentation task and treat unlabeled organs as the background, which may be misleading since the organ unlabeled in this dataset is indeed the foreground on another task. Moreover, most of these methods adopt the multi-head architecture, which is composed of a shared backbone network and multiple segmentation heads for different tasks. Each head is either a decoder [2] or the last segmentation layer [21]. In the paper, we propose a hybrid supervised framework, StMt, that integrates self-training and mean teacher for the segmentation of abdominal organs and tumors using partially labeled and unlabeled data. We introduce a two-stage segmentation pipeline and whole-volume-based input strategy to maximize segmentation accuracy while meeting the requirements of inference time and GPU memory usage.

## 2   Method

We conducted an analysis of the distribution of labels in the labeled data, as depicted in Figure 1. We define datasets that include labels for all 13 organs

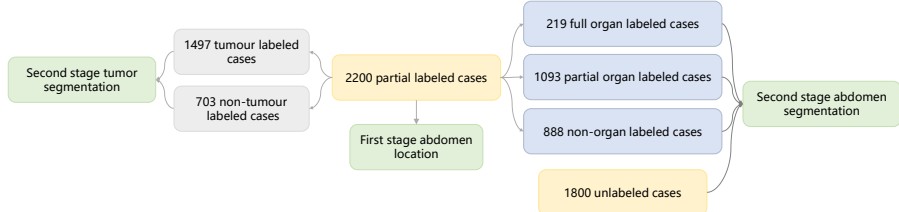

**Fig. 1.** The statistics and utilization of partial=labeled data and unlabeled data.

as 'fully organ labeled cases'. Those without any organ annotations are termed 'non-organ labeled data', and data with annotations for some but not all of the 13 organs are referred to as 'partially labeled organ Data'. Similarly, the data is categorized based on the presence or absence of tumors into two distinct groups: 'Tumor-Annotated Data' and 'Non-Tumor-Annotated Data'. Specifically focusing on abdominal organs, we found that out of a total of 219 cases, all 13 organs were fully annotated. Moreover, there were 1093 cases with partial annotations, indicating that only specific organ categories were annotated. The remaining 888 cases had no annotations. For tumors, 1497 cases have annotations, and the remaining 703 cases do not. It is worth noting that within the annotated cases, there may still be unlabeled regions that potentially contain tumors. We introduce a two-stage segmentation pipeline [14] to maximize segmentation accuracy while meeting the requirements of inference time and GPU memory usage, in where the first-stage aims to obtain the rough location of the abdomen and the second-stage achieves precise segmentation of abdominal organs and tumour based on the first-stage location. Considering the uncertainty in the distribution of tumors, we divided second-stage segmentation task into two subtasks: semi-supervision organ segmentation and tumor segmentation. The method is described in detail in the following subsections.

### 2.1   Preprocessing

The preprocessing strategy for input data in the two-stage segmentation framework is as follows:

- Resampling images to uniform sizes. We use small-scale images as the input of the two-stage segmentation to improve the segmentation efficiency. First-stage input: [128, 128, 128]; Second-stage input: [192, 192, 192].
- We uses the 0.5 and 99.5 percentiles of the foreground voxels for clipping as well as the global foreground mean and s.d. for the normalization of all images [12].
- In training phase, considering that the purpose of first-stage segmentation is to roughly extract the locations of abdomen, we set the voxels whose intensity values are greater than 1 in the resampled ground truth to 1, which converts the multi-classification abdominal organ and tumour segmentation

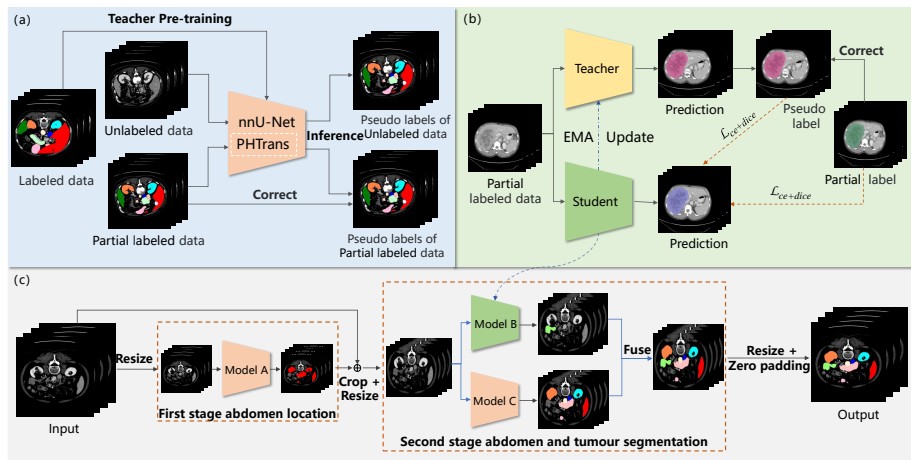

**Fig. 2.** (a) Self-training with partially annotated data and unlabeled data. (b) Tumor segmentation based on Mean Teacher with hybrid supervision. (c) Two-stage inference pipeline for abdominal organ and tumor segmentation.

into a simple two-classification integrated abdomen segmentation. Furthermore, we set the label of tumors in the input data for the second-stage organ segmentation as 0, while setting the label of organs in the input data for tumor segmentation as 0.

## 2.2   Proposed Method

We follow self-training to segment abdominal organs in both the first and second stages: 1) train a teacher using fully annotated abdominal organ data, 2) generate pseudo-labels for partial labeled and unlabeled data, 3) train a student using both labeled data and pseudo-labeled data. In order to obtain high-accuracy pseudo-labels, PHTrans [13], a hybrid network consisting of CNN and Swin Transformer, replaces U-Net as the network within the nnU-Net framework for training the teacher and generating generate precise pseudo labels for partial labeled data and unlabeled data. The student model employ a smaller Res-UNet to reduce memory consumption and utilizes a whole-volume-based input strategy to improve inference efficiency. As shown in the Figure 2 (a), in step 2), we use partial annotations to correct pseudo-labels. Specifically, we calculate the category set A of organ annotations in the partial labels. We set the voxels in the pseudo-labels with values belonging to set A to 0. Finally, we assign the same label to the pseudo-labels based on the position indices of the annotated voxels in the partial labels. As a result, we obtained three types of labeled data: fully labeled cases, corrected pseudo-labeled (CPL) cases, and pseudo-labeled (PL) cases. We fed them into the student model, where the input batch for model training consisted of these three types of data in equal proportions. We calculated the loss function for each of these three types of data separately. Finally,

the training objective $\mathcal{L}_o$ of organ segmentation is formulated as

$$\mathcal{L}_o = \mathcal{L}_{ol} + \lambda_1 \mathcal{L}_{cpl} + \lambda_2 \mathcal{L}_{pl} \tag{1}$$

where, $\lambda_1$ and $\lambda_2$ are the weight of loss components $\mathcal{L}_{cpl}$ and $\mathcal{L}_{pl}$.

The tumor segmentation task cannot follow traditional SSS settings due to the possibility of tumors being unannotated in partial labels. Pan-cancer Segmentation of the abdomen includes various types such as liver cancer and kidney cancer. Each annotated case only contains some types of tumors, and there may be unannotated tumors. As shown in the Figure 2 (b), we follow the idea of model weight aggregation in the mean teacher approach. The teacher model utilizes Exponential Moving Average (EMA) on the student model to update itself, aggregating all the previously learned representation information. Thanks to the updating mechanism of the teacher model, the model can explore the semantic representation of potential unannotated tumors. Therefore, in each training iteration, we use the predictions generated by the teacher model as pseudo-labels to provide additional supervision information. Specifically, similar to the pseudo-labels for organ segmentation, we make real-time corrections to the pseudo-labels. The training objective $\mathcal{L}_t$ of tumour segmentation is formulated as

$$\mathcal{L}_t = \mathcal{L}_{tl} + \lambda \mathcal{L}_{cpl} \tag{2}$$

where, $\lambda$ is the weight of loss components $\mathcal{L}_{cpl}$.

Similarly, both the teacher and the student for tumor segmentation employ a smaller Res-UNet and a whole-volume-based input strategy to improve inference efficiency. Due to the poor performance of the tumor segmentation model trained using labeled data, unlabeled images were not used. The objective of each model training is to minimize the composite loss function, which is a combination of dice loss and cross-entropy loss. Moreover, We not used the pseudo labels generated by the FLARE22 winning algorithm [11] and the best-accuracy-algorithm [24]. As shown in the Figure 2 (c), in inference phase, the segmentation model of first-stage obtain the rough location of the abdomen from the whole CT volume. The second-stage achieves precise segmentation of abdominal organs and tumour based on cropped ROIs from the first-stage segmentation result. Then, the results of abdominal organ segmentation and tumor segmentation are merged, i.e., overlaying the segmented tumor onto the organ segmentation results. Finally, the result is restored to the size of the original data by resampling and zero padding.

## 2.3 Post-processing

Connected component-based post-processing is commonly used in medical image segmentation. Especially in organ image segmentation, it often helps to eliminate the detection of spurious false positives by removing all but the largest connected component. We applied it to the output of the second-stage organ segmentation.

## 3   Experiments

### 3.1   Dataset and evaluation measures

The FLARE 2023 challenge is an extension of the FLARE 2021-2022 [16][17], aiming to aim to promote the development of foundation models in abdominal disease analysis. The segmentation targets cover 13 organs and various abdominal lesions. The training dataset is curated from more than 30 medical centers under the license permission, including TCIA [4], LiTS [1], MSD [23], KiTS [9,10], autoPET [8,7], TotalSegmentator [25], and AbdomenCT-1K [18]. The training set includes 4000 abdomen CT scans where 2200 CT scans with partial labels and 1800 CT scans without labels. The validation and testing sets include 100 and 400 CT scans, respectively, which cover various abdominal cancer types, such as liver cancer, kidney cancer, pancreas cancer, colon cancer, gastric cancer, and so on. The organ annotation process used ITK-SNAP [27], nnU-Net [12], and MedSAM [15].

The evaluation metrics encompass two accuracy measures—Dice Similarity Coefficient (DSC) and Normalized Surface Dice (NSD)—alongside two efficiency measures—running time and area under the GPU memory-time curve. These metrics collectively contribute to the ranking computation. Furthermore, the running time and GPU memory consumption are considered within tolerances of 15 seconds and 4 GB, respectively.

### 3.2   Implementation details

The training and inference of the teacher model in self-training utilize the default configuration of nnU-Net. For organ segmentation, the loss weights $\lambda_1$ and $\lambda_2$ for the student model are set as 1 and 0.5, respectively. The loss weight $\lambda$ for correcting pseudo labels in tumor segmentation is set to 1. To achieve model lightweight, the base number of channels of Res-UNet is set to 16 and the number of up-sampling and down-sampling is 5. The development environments and requirements are presented in Table 1. The training protocols for first-stage and second-stage segmentation are presented in Table 2. To alleviate the over-fitting of limited training data, we employed online data argumentation, including random rotation, scaling, adding white Gaussian noise, Gaussian blurring, adjusting rightness and contrast, simulation of low resolution, Gamma transformation, and elastic deformation.

## 4   Results and discussion

### 4.1   Quantitative results on validation set.

We used the default nnU-Net with full supervision to train on 2200 labeled data as the baseline. We conducted the following experiments: 1) two-stage segmentation experiments using the **F**ully **S**upervised method on 219 full **O**rgans labeled data (FSO); 2) two-stage segmentation experiments using the self-training

**Table 1.** Development environments and requirements.

| System | Ubuntu 20.04.3 LTS |
|---|---|
| CPU | AMD EPYC 7742 64-Core Processor |
| RAM | 94 GB; 2933 MT/s |
| GPU (number and type) | One NVIDIA A100 80G |
| CUDA version | 11.6 |
| Programming language | Python 3.10.11 |
| Deep learning framework | Pytorch (Torch 2.0.0, torchvision 0.15.0) |
| Specific dependencies | nnU-Net |

**Table 2.** The training protocols of two-stage segmentation framework.

| Model | first-stage model / organ seg model / tumour seg model |
|---|---|
| Batch size | 2 / 3 / 2 |
| Patch size | 128×128×128 / 192×192×192 / 192×192×192 |
| Total epochs | 500 |
| Optimizer | SGD with nesterov momentum ($\mu = 0.99$) |
| Initial learning rate (lr) | 0.01 |
| Lr decay schedule | Poly LR |
| Training time (hours) | 17.28 / 47.88/ 31 |
| Number of model parameters | 45.87 M[4] |
| Number of flops | 372.55 / 1886.03/ 1257.35G[5] |
| $CO_2$eq | 1.40 / 7.27 / 5.00 kg[6] |

**Table 3.** Ablation studies. (FSO: two-stage segmentation experiments using the Fully Supervised method on 219 full Organs labeled data; FST: two-stage segmentation experiments using the Fully Supervised method on 1497 Tumor labeled data.)

| Methods | Organ | | Tumour | |
|---|---|---|---|---|
| | DSC(%) | NSD(%) | DSC(%) | NSD(%) |
| nnU-Net | 38.76 | 40.32 | 46.8 | 35.47 |
| FSO | 87.86 | 94.27 | — | — |
| Self-training with part label | 89.1 | 95.58 | — | — |
| Self-training with part label and unlabel | 89.6 | 96.19 | — | — |
| FST | — | — | 46.69 | 39.02 |
| Mean teacher | — | — | 52.08 | 42.82 |
| StMt | 89.6 | 96.19 | 52.08 | 42.82 |

method on 219 full organs labeled data and 1093 partial labeled data; 3) two-stage segmentation experiments introducing 888 + 1800 unlabeled data based on experiment 2); 4) two-stage segmentation experiments using the **F**ully **S**upervised method on 1497 **T**umor labeled data (FST); 5) two-stage segmentation experiments using mean teacher and pseudo-label supervision based on experiment 4); 6) hybrid supervision of **S**elf-**t**raining and **M**ean **t**eacher for organ-tumor segmentation (StMt), which is the combination of experiments 3) and 5).

Table 3 shows that the average organ DSC on the validation set of nnU-Net is only 38.76%. The reason for this result is that when training partial labeled data in a fully supervised manner, the unlabeled organs are considered as background, which can lead to ambiguity during training optimization and result in lower performance. With FSO training using only full organ labeled data, the DSC for organs is 87.86%. By introducing pseudo-label data from partial labeled data through self-training, the DSC improves to 89.1%. Furthermore, with the additional introduction of pseudo-labels from unlabeled data, the DSC further increases to 89.6%. The NSD exhibits a similar changing trend. These results fully demonstrate that both partial label and unlabeled data are beneficial for performance improvement. Using Mean teacher for tumor segmentation has shown improvements of 5.39% in DSC and 3.8% in NSD compared to FST. StMt integrates self-training and mean teacher to achieve the best segmentation results. Table 4 presents detailed results for StMt in terms of public Validation, online validation, and test submission.

**Table 4.** Quantitative evaluation results.

| Target | Public Validation | | Online Validation | | Testing | |
|---|---|---|---|---|---|---|
| | DSC(%) | NSD(%) | DSC(%) | NSD(%) | DSC(%) | NSD (%) |
| Liver | $97.41 \pm 0.60$ | $99.32 \pm 0.51$ | 97.40 | 99.21 | 96.33 | 97.94 |
| Right Kidney | $94.18 \pm 6.14$ | $95.74 \pm 8.13$ | 93.78 | 95.51 | 93.60 | 94.62 |
| Spleen | $95.29 \pm 1.93$ | $97.94 \pm 2.99$ | 95.78 | 98.45 | 95.37 | 98.04 |
| Pancreas | $85.95 \pm 5.22$ | $97.50 \pm 3.71$ | 85.42 | 97.01 | 89.05 | 98.15 |
| Aorta | $94.87 \pm 0.97$ | $99.09 \pm 1.13$ | 94.77 | 98.91 | 95.31 | 99.68 |
| Inferior vena cava | $92.81 \pm 2.09$ | $97.38 \pm 2.14$ | 92.86 | 97.35 | 93.21 | 98.05 |
| Right adrenal gland | $82.76 \pm 5.03$ | $96.98 \pm 2.47$ | 81.67 | 96.50 | 81.19 | 95.21 |
| Left adrenal gland | $81.54 \pm 5.14$ | $96.26 \pm 3.37$ | 80.91 | 95.31 | 81.36 | 94.38 |
| Gallbladder | $87.12 \pm 18.85$ | $89.15 \pm 19.90$ | 89.92 | 91.40 | 87.26 | 90.44 |
| Esophagus | $92.39 \pm 14.93$ | $93.29 \pm 14.72$ | 83.56 | 94.76 | 89.53 | 98.89 |
| Stomach | $93.40 \pm 5.11$ | $97.49 \pm 5.68$ | 93.98 | 98.03 | 94.53 | 98.10 |
| Duodenum | $83.60 \pm 6.68$ | $95.69 \pm 4.75$ | 84.72 | 96.27 | 88.40 | 97.87 |
| Left kidney | $93.49 \pm 6.24$ | $94.68 \pm 8.91$ | 92.55 | 94.40 | 92.26 | 93.97 |
| Tumor | $52.08 \pm 34.09$ | $42.82 \pm 28.69$ | 45.55 | 37.82 | 50.00 | 38.32 |
| Average | $86.92 \pm 8.07$ | $92.38 \pm 7.65$ | 86.63 | 92.21 | 87.59 | 92.35 |

## 4.2   Qualitative results on validation set

We visualize the segmentation results of the validation set. The representative samples in Figure 3 demonstrate the success of identifying organ details by StMt, which is the closest to the ground truth compared to other methods due to retaining most of the spatial information of abdominal organs and tumour. In particular, it outperforms FSO significantly by leveraging partial labeled and unlabeled data with self-training, which enhances the generalization of the segmentation model. Compared to FST, mean Teacher achieves more complete tumor segmentation. Furthermore, we show representative examples of poor segmentation. The third row demonstrates that none of the methods were able to segment the tumor (atrovirens region) in the lower abdomen. Due to the fact that the 13 organ classes in the Flare2023 dataset are primarily focused on the upper abdomen, it is difficult to accurately locate the approximate position of the abdominal area containing the tumor in the first stage by relying solely on the tumor. The fourth row shows another case where none of the methods accurately detected the spleen (blue region).

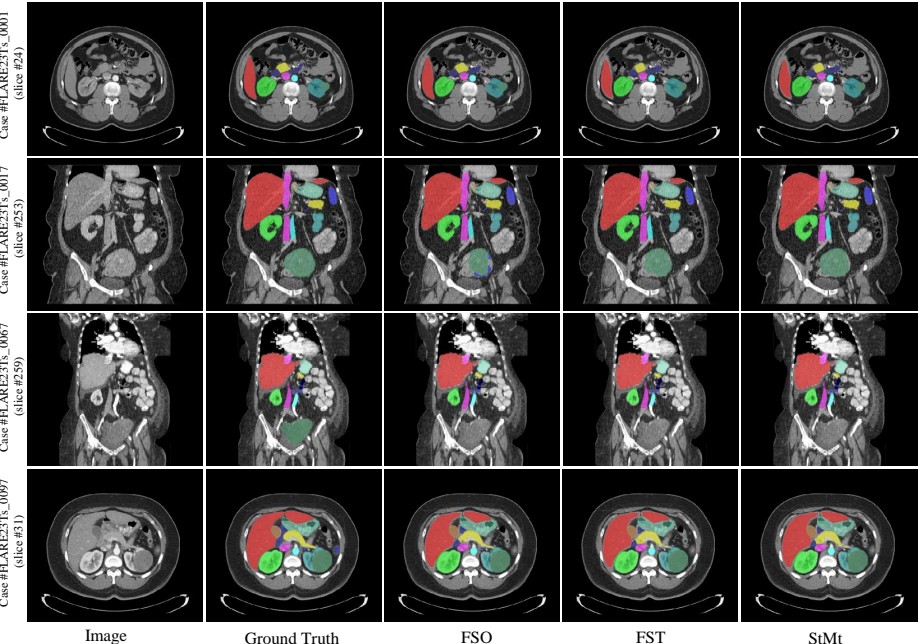

**Fig. 3.** Visualization of segmentation results of abdominal organs and tumour.

### 4.3   Segmentation efficiency results on validation set

In the official segmentation efficiency evaluation, the average inference time of 100 cases in the validation set is 11.25 s, the average maximum GPU memory is 3519.06 MB, and the area under the GPU memory-time curve is 9627.82MB. Thanks to the two-stage framework and the whole-volume-based input strategy, the inference time for each test case is within 15 seconds. Additionally, the small Res-UNet and input size ensure that the GPU memory usage remains below 4GB. The detailed quantitative results of the segmentation efficiency for some cases are shown in Table 5.

**Table 5.** Quantitative evaluation of segmentation efficiency in terms of the running them and GPU memory consumption. Total GPU denotes the area under GPU Memory-Time curve. Evaluation GPU platform: NVIDIA QUADRO RTX5000 (16G).

| Case ID | Image Size | Running Time (s) | Max GPU (MB) | Total GPU (MB) |
|---------|------------|------------------|--------------|----------------|
| 0001 | (512, 512, 55) | 11.76 | 3220 | 10849 |
| 0051 | (512, 512, 100) | 10.2 | 3044 | 8833 |
| 0017 | (512, 512, 150) | 10.62 | 3200 | 9204 |
| 0019 | (512, 512, 215) | 11.14 | 3800 | 9557 |
| 0099 | (512, 512, 334) | 11.92 | 3800 | 9941 |
| 0063 | (512, 512, 448) | 13.51 | 3582 | 10614 |
| 0048 | (512, 512, 499) | 13.85 | 3800 | 10724 |
| 0029 | (512, 512, 554) | 14.6 | 3800 | 11134 |

### 4.4   Results on final testing set

In the testing set, the StMt model achieved average DSC scores of 90.48% for organs and 50.00% for lesions, along with NSD scores averaging 96.51% for organs and 38.32% for lesions. Additionally, the average running time was 11.1 seconds, and the area under the GPU memory-time curve was 8979 MB.

### 4.5   Limitation and future work

Due to the limited time available for participating in the challenge, our work still has many shortcomings. For example, the segmentation performance of tumors is poor, partly due to the fact that the tumor in the lower abdomen was not successfully segmented in the first stage. It is possible to try performing tumor segmentation independently in the first stage and then integrating the results with organ segmentation, similar to the second stage. Furthermore, selecting high-quality pseudo-labeled data may contribute to improving segmentation performance, and it is worth a try.

## 5 Conclusion

In the paper, we propose a hybrid supervised framework, StMt, that integrates self-training and mean teacher for the segmentation of abdominal organs and tumors using partially labeled and unlabeled data. We introduce a two-stage segmentation pipeline and whole-volume-based input strategy to maximize segmentation accuracy while meeting the requirements of inference time and GPU memory usage. Experiments on the validation set of FLARE2023 demonstrate that our method achieves excellent segmentation performance as well as fast and low-resource model inference. Our method achieved an average DSC score of 89.79% and 45.55 % for the organs and lesions on the validation set and the average running time and area under GPU memory-time cure are 11.25s and 9627.82MB, respectively.

**Acknowledgements** The authors of this paper declare that the segmentation method they implemented for participation in the FLARE 2023 challenge has not used any pre-trained models nor additional datasets other than those provided by the organizers. The proposed solution is fully automatic without any manual intervention. We thank all the data owners for making the CT scans publicly available and CodaLab [20] for hosting the challenge platform.

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

**Table 6.** Checklist Table. Please fill out this checklist table in the answer column.

| Requirements | Answer |
| --- | --- |
| A meaningful title | Yes |
| The number of authors ($\leq 6$) | 6 |
| Author affiliations and ORCID | Yes |
| Corresponding author email is presented | Yes |
| Validation scores are presented in the abstract | Yes |
| Introduction includes at least three parts: background, related work, and motivation | Yes |
| A pipeline/network figure is provided | Figure 2 |
| Pre-processing | Page 3 |
| Strategies to use the partial label | Page 4 |
| Strategies to use the unlabeled images. | Page 4 |
| Strategies to improve model inference | Page 3 |
| Post-processing | Page 5 |
| Dataset and evaluation metric section is presented | Page 5 |
| Environment setting table is provided | Table 1 |
| Training protocol table is provided | Table 2 |
| Ablation study | Page 7 |
| Efficiency evaluation results are provided | Table 5 |
| Visualized segmentation example is provided | Figure 3 |
| Limitation and future work are presented | Yes |
| Reference format is consistent. | Yes |