# OpenReview forum: "Two-Stage Hybrid Supervision Framework for Fast, Low-resource, and Accurate Organ and Pan-cancer Segmentation in Abdomen CT"
_MICCAI.org/2023/FLARE — Submitted to FLARE 2023_

### Official Review · Reviewer_D5hE · 2023-10-03
**Two-Stage Hybrid Supervision Framework for Fast, Low-resource, and Accurate Organ and Pan-cancer Segmentation in Abdomen CT**

**Rating:** 8
**Confidence:** 4

**Review:**

This article introduces a hybrid supervised framework called StMt, which integrates self-training and mean teacher methods. It utilizes both partially labeled and unlabeled data for abdominal organ and tumor segmentation. The segmentation performance is excellent, with an average DSC score of 89.79% for organs and 45.55% for lesions on the validation set. The average GPU runtime is also only 11.25 seconds.

Cons:

1. ORCID for multiple authors has not been annotated.

2.The Checklist Table MUST table is not included.

---

> ### Author Response · Authors · 2023-11-14
>
> Comment:
>
> 1. ORCID has been added.
>
> 2. The checklist table has been added.

---

> > ### Comment · Reviewer_D5hE · 2023-11-30
> >
> > Thank you for responding to my suggestions and making changes. I also wish you success in your future work.

---

### Official Review · Reviewer_jZ8j · 2023-10-04
**Review for "Two-Stage Hybrid Supervision Framework for Fast, Low-resource, and Accurate Organ and Pan-cancer Segmentation in Abdomen CT"**

**Rating:** 7
**Confidence:** 4

**Review:**

The article is well-structured and introduce a two-stage segmentation pipeline and whole-volume-based input strategy to maximize segmentation accuracy.

Cons:
1. The ORCIDs for authors are not been included.
2. In Introduction, the paper didn't include the related work/state-of-the-art methods on semi-supervised/partial-label segmentation.
3. The words in Fig.3 are not in correct format.

---

> ### Author Response · Authors · 2023-11-14
>
> Comment:
>
>
> 1. ORCID has been added.
>
>
> 2. The work related to semi-supervised/partial label segmentation has been added to the introduction.
>
> Typically, semi-supervised segmentation (SSS) can be employed to resolve this issue. SSS aims to explore tremendous unlabeled data with supervision from limited labeled data. Recently, self-training methods have dominated this field. Furthermore, methods employing consistency regularization strategies improve the generalization ability by encouraging high similarity in predictions from two perturbed networks for the same input image. In this challenge, due to the fact that the annotation data only includes annotations for partial organs or tumors, traditional SSS methods struggle to achieve excellent segmentation results. The key to developing segmentation algorithms lies in fully leveraging the semantic representation in partially labeled data and extending it to unlabeled cases to enhance the algorithm's generalization. Segmentation of multiple organs and tumors is a generally recognized difficulty in medical image analysis~\cite{dodnet}, particularly when there is no large-scale fully labeled datasets. To address this issue,some works formulate the partially labeled issue as a multi-class segmentation task and treat unlabeled organs as the background, which may be misleading since the organ unlabeled in this dataset is indeed the foreground on another task. Moreover, most of these methods adopt the multi-head architecture, which is composed of a shared backbone network and multiple segmentation heads for different tasks. Each head is either a decoder or the last segmentation layer.
>
> 3. The font of words in Figure 3 has been corrected

---

> > ### Comment · Reviewer_jZ8j · 2023-11-30
> > **2nd round Review**
> >
> > In terms of completeness, this article is currently well written.

---

### Official Review · Reviewer_JKnw · 2023-10-04
**Two-Stage Hybrid Supervision Framework for Fast, Low-resource, and Accurate Organ and Pan-cancer Segmentation in Abdomen CT**

**Rating:** 7
**Confidence:** 4

**Review:**

This paper introduces StMt, a hybrid supervised framework that appropriately utilizes self-training and mean teacher method. The provided dataset is appropriately segmented for different purposes, and the two-stage segmentation consists of learning strategies suitable for each stage.

- pros
    - The 45.55% lesion dsc and fast inference speed of 11.25s is very impressive.
    - It's impressive that they experimented with different models, had clear goals for each learning step, and used the right data.

- cons
    - There is no clear way to classify the statistics of the partially labeled and unlabeled data corresponding to Fig. 1.
    - Article does not contain an ORCID

---

> ### Author Response · Authors · 2023-11-14
>
> Comment:
>
> 1. The details of statistical classification of partially labeled and unlabeled data are as follows:
>
> We conducted an analysis of the distribution of labels in the labeled data, as depicted in Figure 1. We define datasets that include labels for all 13 organs as fully organ labeled cases. Those without any organ annotations are termed non-organ labeled data, and data with annotations for some but not all of the 13 organs are referred to as partially labeled organ Data. Similarly, the data is categorized based on the presence or absence of tumors into two distinct groups: Tumor-Annotated Data and Non-Tumor-Annotated Data. Specifically focusing on abdominal organs, we found that out of a total of 219 cases, all 13 organs were fully annotated. Moreover, there were 1093 cases with partial annotations, indicating that only specific organ categories were annotated. The remaining 888 cases had no annotations. For tumors, 1497 cases have annotations, and the remaining 703 cases do not.
>
> 2. ORCID has been added.

---

### Official Review · Reviewer_gZER · 2023-10-04
**This study uses a two-stage segmentation framework, sampling self-training and average teacher strategy to improve the segmentation of the model. In addition, the input of the model is based on a full-volume image to meet time and memory requirements, achieving an average DSC score of 89.79% for organs and 45.55% for tumours on the online validation leaderboard.**

**Rating:** 8
**Confidence:** 5

**Review:**

This paper is well organized and shows sufficient details of the proposed method. The proposed method still has some potential in tumour segmentation tasks. However, the font in the picture is too small, and the method of abdominal organs in the picture does not conform to the habit of reading, that is, the direction is opposite (Fig. 2). In addition, there is a lack of checklist table in this article.

---

> ### Author Response · Authors · 2023-11-14
>
> Comment:
>
> 1. The font size in the graph has been changed.
>
> 2. The checklist table has been added.

---

> > ### Comment · Reviewer_gZER · 2023-11-30
> >
> > Perhaps what I stated before was not so clear, the CT image with inverted orientation is still preserved in Figure 2 of the current revised paper. Of course, this does not affect the understanding of the paper, but we will feel that it is not quite in line with the habit of browsing medical images, especially for people who read medical images regularly.

---

### Decision · Program_Chairs · 2023-10-24

Accept